# The Plasticity of Pancreatic β-Cells

**DOI:** 10.3390/metabo11040218

**Published:** 2021-04-02

**Authors:** Norikiyo Honzawa, Kei Fujimoto

**Affiliations:** 1Division of Diabetes, Metabolism and Endocrinology, Department of Internal Medicine, Jikei University School of Medicine, 3-25-8, Nishishinbashi, Minato-ku, Tokyo 105-8461, Japan; mamnoon.am.m.m.h.m@jikei.ac.jp; 2Division of Diabetes, Metabolism and Endocrinology, Department of Internal Medicine, Jikei University Kashiwa Hospital, 163-1, Kashiwashita, Kshiwa-shi, Chiba 277-8567, Japan

**Keywords:** pancreatic β cell, dedifferentiation, transdifferentiation

## Abstract

Type 2 diabetes is caused by impaired insulin secretion and/or insulin resistance. Loss of pancreatic β-cell mass detected in human diabetic patients has been considered to be a major cause of impaired insulin secretion. Additionally, apoptosis is found in pancreatic β-cells; β-cell mass loss is induced when cell death exceeds proliferation. Recently, however, β-cell dedifferentiation to pancreatic endocrine progenitor cells and β-cell transdifferentiation to α-cell was reported in human islets, which led to a new underlying molecular mechanism. Hyperglycemia inhibits nuclear translocation and expression of forkhead box-O1 (FoxO1) and induces the expression of neurogenin-3 (Ngn3), which is required for the development and maintenance of pancreatic endocrine progenitor cells. This new hypothesis (Foxology) is attracting attention because it explains molecular mechanism(s) underlying β-cell plasticity. The lineage tracing technique revealed that the contribution of dedifferentiation is higher than that of β-cell apoptosis retaining to β-cell mass loss. In addition, islet cells transdifferentiate each other, such as transdifferentiation of pancreatic β-cell to α-cell and vice versa. Islet cells can exhibit plasticity, and they may have the ability to redifferentiate into any cell type. This review describes recent findings in the dedifferentiation and transdifferentiation of β-cells. We outline novel treatment(s) for diabetes targeting islet cell plasticity.

## 1. Introduction

Blood glucose levels in the body are regulated by hormones secreted by pancreatic endocrine cells. More than 95% of the human pancreas is considered to be made up of exocrine tissue, and 3–5% of it is made up of pancreatic islets, which are an endocrine tissue. The human pancreas contains about 3.2–14.8 million islets, with a total volume of 0.5–2 cm^2^ [1]. Pancreatic β-cells account for approximately 60% of the islets in humans (pancreatic α-cells account for approximately 30%), whereas in rodents, they account for about 80% [1]. The islets are made up of β-cells as well as α, δ, PP, and ε cells that secrete hormones such as insulin, glucagon, somatostatin, pancreatic polypeptide and ghrelin, respectively. Blood glucose is mainly regulated by insulin, glucagon and somatostatin to maintain blood glucose homeostasis. Type 2 diabetes is caused by abnormal secretion of these hormones, which are related to changes in the number and mass of islet cells.

Pancreatic β-cell function and mass increase in the early stage of type 2 diabetes due to insulin resistance, but β-cell function and mass are reduced eventually [2]. The amounts of β-cells in autopsies of type 2 diabetes patients have been estimated to be about 24–65% of that in healthy individuals [3]. Earlier, the main cause of decreased pancreatic β-cell function/mass was thought to be due to pancreatic β-cell apoptosis. Pancreatic β-cell apoptosis can be explained with the help of glucose toxicity, lipotoxicity, amyloid deposition, and/or senescence. Although the detailed mechanism of pancreatic β-cell damage due to glucotoxicity is described below; briefly, chronic hyperglycemia induces oxidative stress or endoplasmic reticulum (ER) stress, leading to the death of pancreatic β-cells [4]. Chronic lipotoxicity leads to pancreatic β-cell dysfunction due to impaired intracellular pathways including oxidative stress, ER stress, autophagy, and ceramide/lipid droplet formation [5]. Moreover, diabetes is a protein conformational disease (PCD) and the oxidative stress associated with hyperglycemia causes abnormal protein accumulation, leading to pancreatic β-cell dysfunction [6]. Islet amyloid polypeptide (IAPP, amylin) is secreted from pancreatic β-cells together with insulin, and along with hyperglycemia its expression increases, leading to extracellular deposition and damage to pancreatic β-cells [7]. Moreover, senescence is accelerated due to an increase in insulin resistance and an increase in cell cycle inhibitors, such as p21^Cis1^ and p16^Ink4a^, is seen with aging. The administration of senolytic compounds to senescent cell-depleted mice or senescence-induced mice using insulin receptor antagonists has been shown to reduce the expression of senescence markers [8]. Thus, to date, apoptosis can explain the cause of pancreatic β-cell dysfunction. However, in recent years, a new concept suggested that the main cause of β-cell (insulin-positive cell) loss is dedifferentiation, and not apoptosis or proliferative disorders [9,10]. In the β-cell-specific FoxO1 knockout mice, a decrease in pancreatic β-cells and an increase in α-cells was observed under metabolic stress conditions such as aging or high fertility, resulting in hyperglycemia [10]. In addition, lineage tracing of β-cell fate in this murine model showed that β-cells changed from insulin-positive cells to insulin-negative cells (empty β-cells). Furthermore, β-cells expressed the transcription factor Ngn3, a pancreatic endocrine progenitor cell marker. This suggests that β-cells were dedifferentiated [10]. In addition, some β-cells turned into pancreatic α-cells, which may explain the pathogenesis of impaired insulin secretion and abnormal pancreatic β/α cell ratios in diabetic patients. Currently, factors that promote or inhibit β-cell dedifferentiation are being researched aggressively. Targeting these dedifferentiation-related factors may contribute to pancreatic β-cell protection and apply to a novel diabetes treatment strategy. In recent years, gene therapy using transdifferentiation and redifferentiation has been reported in mice [11]; such an approach holds promise in the development of targeting α-β transdifferentiation. In this review, an outline of pancreatic β-cell dysfunction and dedifferentiation due to glucotoxicity and the potential for therapies are given.

## 2. β-Cells Dysfunction via Oxidative Stress or ER Stress

β-cell dysfunction in type 2 diabetes suggests decreased insulin synthesis and secretion. Chronic hyperglycemia leads to activation of protein glycation and a mitochondrial electron transfer system, resulting in oxidative stress. β-cells are particularly vulnerable to oxidative stress due to their low antioxidant enzyme expression levels and activity [12]. Oxidative stress reduces MafA and pancreas duodenal homeobox gene 1 (Pdx-1) expression, which are pivotal transcription factors for β-cell development and maintenance, resulting in decreased insulin biosynthesis and secretion [13,14]. Treating obese type 2 diabetes model mice with antioxidants showed a protective effect on β-cells; this effect was also seen in and the HIT-T15 β-cell line [15]. It has also reported that FoxO1, a transcription factor involved in insulin signaling, suppresses oxidative stress and protects β-cells via MafA [16].

Pdx-1 was thought to be important for β-cell survival, cell death and differentiation [17]. In a hyperglycemic state, c-Jun N-terminal kinase (JNK) activation is increased due to oxidative stress, which in turn, reduces Pdx-1 expression, and results in decreased insulin expression and impaired secretion [18]. Furthermore, β-cell-specific Pdx-1 overexpression in diabetic mouse models restores β-cell function and improves glycemic control [19], suggesting that Pdx-1 is involved in β-cell dysfunction. MafA is also involved in β-cell function; increased expression of c-Jun and decreased expression of MafA and insulin are observed in the pancreatic islets of diabetic model mice [20]. Overexpression of c-Jun in the MIN6 β-cell line as well as in isolated mouse islets results in decreased expression of MafA and insulin [20]. Furthermore, MafA overexpression in the β-cells of obese type 2 diabetic model mice recovered their ability to synthesize insulin, and improved glucose-stimulated insulin secretion (GSIS) as well as their blood glucose levels [21]. These results indicate that c-Jun expression regulates MafA, and is involved in β-cell dysfunction. In addition, knockdown of MafA in β-cells results in decreased expression of Pdx-1, indicating a close relationship between MafA and Pdx-1 [22]. Thus, chronic hyperglycemia in β-cells increases oxidative stress and causes β-cell dysfunction due to decreased MafA and Pdx-1.

Furthermore, hyperglycemia induces ER stress by causing the accumulation of poorly folded/unfolded proteins (e.g., misfolded-proinsulin, IAPP) in pancreatic β-cells [23]. In the ER, unfolded protein responses (UPR) to the accumulation of these unfolded proteins activate working membrane proteins in the ER, such as dsRNA-activated protein kinase (PKR)-like ER kinase (PERK), inositol requiring protein 1α (IRE1α), and activating transcription factor 6 (ATF6), thereby reducing the load on the ER [4]. This increases the amount of chaperone proteins involved in protein folding that then attempts to remove unfolded proteins. When UPR cannot deal with the ER stress, this results in pancreatic β-cell apoptosis [24,25]. Chronic ER stress induces numerous apoptotic signals including oxidative stress, IRE1α-mediated activation of apoptosis signal-regulating kinase 1/c-Jun amino-terminal kinase (ASK1/JNK), PERK-dependent C/EBP homologous protein (CHOP) expression, and activation of caspase-12, caspase-3, and endogenous mitochondria-dependent cell death pathways [4]. In this manner, chronic hyperglycemia induces oxidative stress and ER stress, leading to pancreatic β-cell dysfunction and cell death (Figure 1).

## 3. Pancreatic β-Cell Dedifferentiation and Transdifferentiation

### 3.1. The History of Dedifferentiation

In 2004, Gershengorn et al. proposed pancreatic β-cell dedifferentiation for the first time. Based on their observation that insulin-positive cells disappeared and fibroblast-like precursor cells appeared in human islets. Furthermore, when human pancreatic islets were cultured under serum-free culture conditions, insulin-positive cells and glucagon-positive cells reappeared, which leads to the suggestion the cells are capable of transdifferentiation and redifferentiation [26]. In 2007, Dor et al. used a lineage tracing technique to demonstrate Pdx-1 and glucose transporter type 2 (GLUT2) disappearance from mouse pancreatic islets after 14 days culture, and insulin-positive cell disappearance in the majority of β-cells after 21 days of culture [27]. In 2012, Accili et al. reported pancreatic β-cell dedifferentiation in vivo. When pancreatic β-cell-specific FoxO1 knockout mice were subjected to metabolic stress due to aging or high fertility, decreased β-cell and increased α cell mass, which resulted in hyperglycemia [10]. Furthermore, although β-cell apoptosis has been considered to be the main cause of β-cell dysfunction, in this murine model, dedifferentiation is the main cause of β-cell dysfunction rather than apoptosis [10]. In addition, the same group reported that the number of insulin-positive cells decreased, and pancreatic β-cell dedifferentiation occurred in a mouse model of insulin-resistant type 2 diabetes (db/db mice) [10]. In addition, in 2014, pancreatic β-cell dedifferentiation was also reported in neonatal diabetes models such as pancreatic β-cell-specific mutant K_ATP_ channel-expressing mice and pancreatic β-cell-specific sulfonylurea receptor-deficient mice [28]. In 2016, it was reported from autopsies of the pancreases of patients with type 2 diabetes that β-cell dedifferentiation occurred 3.5 times more in patients with type 2 diabetes than in controls [3]. In addition, a negative correlation was observed between dedifferentiation and the capacity for insulin secretion [3]. Thus, while β-cell apoptosis is one cause of β-cell failure, β-cell dedifferentiation is now considered to be a causative factor for β-cell failure as well [9].

### 3.2. Dedifferentiation Mechanisms

Pancreatic β-cell dedifferentiation shows the conversion of mainly mature β-cells to endocrine progenitor cells. It is defined by (1) the loss of pancreatic mature β-cell markers (such as MafA, Insulin, Nkx6.1 and Pdx-1) and (2) the appearance of endocrine progenitor cell and stem-like cell markers (Ngn3, Oct4, Aldh1a, Nanog, L-Myc and Sox9) [29,30]. Additionally, many studies have examined conditions and molecules that promote/inhibit pancreatic β-cell dedifferentiation.

Hyperglycemia, inflammatory cytokines, Hedgehog (Hh) signaling (which is involved in differentiation and development), the renin–angiotensin–aldosterone system (RAAS, which is involved in regulating blood pressure and circulating blood volume), miRNA (non-coding RNA that regulates gene expression), noradrenaline, and dopamine receptor D2 (DRD2) have been reported to promote pancreatic β-cell dedifferentiation. Hyperglycemia is thought to promote pancreatic β-cell dedifferentiation through oxidative stress (reactive oxygen species (ROS)). Accumulation of ROS in β-cells has been shown to suppress the Pdx1 and MafA transcription factors’ expression and ultimately reduce insulin synthesis [31]. In addition, db/db mice with hyperglycemia had increased levels of pancreatic β-cell dedifferentiation independently of blood lipid levels; their pancreatic islets had decreased levels of mature β-cell markers such as GLUT2, Pdx-1, MafA and Nkx6.1 [32,33]. Thus, it seems to be all but certain that hyperglycemia promotes pancreatic β-cell dedifferentiation. Inflammatory cytokines (IL-1, IL-6, TNF-α) have also been reported to promote pancreatic β-cell dedifferentiation. Exposing human pancreatic islets to IL-1β induces the downregulation of FoxO1, MafA, Nkx6.1, and Pdx-1 mRNA expressions [34]. Additionally, anti-inflammatory treatments such as anti-IL-1β, anti-TNFα, or anti-NF-kB improved β-cell insulin secretion capacity in diabetic model mice [34]. However, on the other hand, pancreas-specific IL-1 knockout mice showed increased Aldh1a3 expression and MafA downregulation, resulting in impaired GSIS [35]. Hh signaling also promotes β-cell dedifferentiation. In mice with pancreas-specific Hh signaling activation, islets showed decreased expression of both mature β-cell markers such as MafA, Nkx6.1, and NeuroD1 and endocrine progenitor cell markers such as Ngn3, and increased expression of stem-like cell markers such as Sox9 and Hes1 [36]. RAAS also induces pancreatic β-cell dedifferentiation and dysfunction via NF-kB activation. Administration of angiotensin II to db/db mice decreased the expression of Pdx-1 and FoxO1 and increased Ngn3 in pancreatic islets. Additionally, when an angiotensin-converting-enzyme (ACE) inhibitor was added, the expression of these transcription factors was maintained [37]. Furthermore, adding an ACE inhibitor to the β-cell line improved the hyperglycemia-induced decrease in Ins1 and Pdx-1 expression, and suppressed the expression of Ngn3, endocrine progenitor cell marker, and Oct4, stem-like cell marker [38]. In addition, pancreatic β-cell dedifferentiation was suppressed by administration of ACE2, which decomposes AT2 and suppresses the RAAS, to mice fed a high-fat diet (HFD). In addition, ACE2 knockout mice showed 20% β-cell dedifferentiation, and increased Oct4 expression [38]. Non-coding RNAs that regulate gene expression such as miR24 and miR302s also promote pancreatic β-cell dedifferentiation. In particular, miR302s, which is expressed in pluripotent/undifferentiated stem cells, suppresses the expression of mature β-cell marker genes (*NeuroD1*, *PPARA*, *DRD1*, and *SLC7A2*) in human pancreatic islets and promotes pancreatic β-cell dedifferentiation [39]. In the same way, miR24 enhances Ngn3 and Sox9 mRNA expression in β-cell lines and Ngn3 expression in pancreatic islet cells; it also suppresses ER stress-induced apoptosis [40]. Recently, noradrenergic neurons have also been implicated in the dedifferentiation and dysfunction of human pancreatic β-cells. Tyrosine hydroxylase (TH)-positive noradrenergic fibers are increased in the pancreatic islets of diabetic patients and positively correlated with their dedifferentiation score (synaptophysin +/4 islet hormones (Ins, Gcg, SST, PP)-) and negatively correlated with insulin secretion [41]. Similarly, dopamine receptor D2 (DRD2) promotes dedifferentiation [42]. The addition of domperidone, an inhibitor of DRD2, to mouse pancreatic islet cells using lineage tracing technology has been shown to reduce dedifferentiated cells (Ins-/EYFP+) [42].

Molecules/peptides such as FoxO1, polycomb repressor complex 2 (PRC2), high mobility group protein 20A (HMG20A), pdx-1 associated long non-cording RNA upregulator of transcription (PLUTO), vesicular monoamine transporter 2 (VMAT2), and glucagon-like peptide-1 (GLP-1) have been reported to inhibit pancreatic β-cell dedifferentiation. FoxO1 inhibits β-cell dedifferentiation, as mentioned above. In pancreatic β-cell-specific *FoxO1*-deficient mice, metabolic stress due to aging or high fertility induced hyperglycemia by decreasing β-cell mass and increasing α-cell mass [10]. Furthermore, lineage tracing of β-cells showed that mature β-cell markers such as Ins, MafA, and Pdx-1 were lost, while endocrine progenior cell markers such as chromogranin A and Ngn3 were detected instead. Further, the expression of stem-like cell markers such as Oct4, Nanog, and L-Myc was induced in dedifferentiated β-cells, suggesting that these dedifferentiated cells exhibit pluripotency [10]. As mentioned above, hyperglycemia decreases FoxO1 expression, suggesting that it is involved in pancreatic β-cell dedifferentiation. Epigenetic mapping using human and mouse single-cell transcriptome has also shown that PRC2, one of two in a set of polycomb repressor complexes in the chromatin regulatory system, is related to β-cells. Pancreatic β-cell-specific PRC2 dysfunction mice have downregulated mature β-cell marker genes (Pdx1, MafA, Nkx2.2, and Nkx6.1) and upregulated immature β-cell and endocrine progenitor cell markers (Atf5, Zfpm1, Gata3, Pax1, etc.) [43]. HMG20A, also known as iBRAF, is a chromatin regulator involved in neuronal differentiation and maturity, and is involved in pancreatic β-cell dedifferentiation. HMG20A is expressed in human and mouse pancreatic islets. Its expression is lower in type 2 diabetic donor islets than in non-diabetic donor islets in humans. Knockdown of HMG20A in an insulin-secreting cell line (INS-1), also caused downregulation of mature β-cell markers such as NeuroD1, MafA, glucokinase (Gck), and Ins, as well as upregulation of pare box gene 4 (Pax4) [44]. Furthermore, PLUTO, a long non-coding RNA (lncRNA) in β-cells, affects the transcription of Pdx-1. PLUTO is down-regulated and suppresses Pdx-1 expression in diabetic patients. Thus, PLUTO may suppress the dedifferentiation of pancreatic β-cells [45,46]. In pancreatic β-cell-specific vesicular monoamine transporter 2 (VMAT2) knockout mice, mature β-cell markers (Ins1, Ins2, MafA, Nkx6.1, Gck, and Pdx-1) decreased with age, and the expression of MafB, an immature β-cell or a mature α cell marker, was upregulated [47]. GLP-1 is also involved in the suppression of pancreatic β-cell dedifferentiation. In pancreatic islet cultures from non-diabetic patients, β-cell dedifferentiation progresses over time, and their insulin content decreases by 50% in about one week. When this happens, the expression of mature β-cell markers (Pdx-1 and MafA) is decreased, and the expression of endocrine progenitor cell markers (Ngn3 and Aldh1a) is increased [48]. The addition of GLP-1 to human islets suppressed the expression of these dedifferentiation markers, and knockdown of FoxO1 diminished these effects, suggesting that FoxO1 may be involved in the GLP-1-induced suppression of pancreatic β-cell dedifferentiation [48].

As mentioned above, the dedifferentiation of pancreatic β-cells has been explained by dividing these into stimulatory and inhibitory factors. The main factor promoting the dedifferentiation of pancreatic β-cells is thought to be oxidative stress signals. Under hyperglycemic conditions, insulin hypersecretion is accompanied with activation of mitochondrial function in pancreatic β-cells, resulting in the production of reactive oxygen species (ROS). ROS causes reduction of MafA and Pdx-1 expressions, transcription factors required for the development and maintenance of pancreatic β-cells, thereby promoting the dedifferentiation of pancreatic β-cells and decreasing insulin secretion [13,14]. In recent years, antioxidant administration has been reported to protect pancreatic β-cells in obese type 2 diabetes model mice and pancreatic β-cell lines (HIT-T15) [14,15] and VMAT2, which is expressed in pancreatic β-cells, has been shown to suppress oxidative stress through modulation of dopamine levels in pancreatic β-cells, resulting in inhibition of pancreatic β-cell death and dedifferentiation [47]. In addition, FoxO1 has also gained attention as an inhibitor of pancreatic β-cell dedifferentiation. At steady-state, FoxO1 is localized in the cytoplasm of pancreatic β-cells; however, it is translocated from the cytoplasm into the nucleus under metabolic stress due to hyperglycemia (including oxidative stress). Thereafter, FoxO1 is proteolyzed and eliminated when hyperglycemia persists [16]. Subsequently, the expression of Ngn3 that is suppressed by FoxO1 is enhanced and pancreatic β-cells undergo dedifferentiation. Furthermore, there is crosstalk between FoxO1 and Notch signaling that leads to decreased expression of the Ngn3 transcriptional repressor, hairy and enhancer of split 1 (Hes1), resulting in dedifferentiation of pancreatic β-cells [49]. Thus, FoxO1 is an important regulator of pancreatic β-cell dedifferentiation and evaluations in humans are anticipated (Figure 2).

### 3.3. Pancreatic α-Cells to β-Cells Transdifferentiation

Dedifferentiation is defined as the transdifferentiation into undifferentiated cells. This leads to worsening blood glucose due to decreased insulin secretion in diabetes. If the loss of β-cell mass can be suppressed, and β-cell regeneration as well as the transdifferentiation of non-β-cells into mature β-cells progresses, then β-cell mass will be maintained or even increased. This is expected to improve diabetes by counteracting hyperglycemia. In other words, the transdifferentiation of non-β-cells into mature β-cells may become a new therapeutic strategy for diabetes. The transdifferentiation of α-cells, which secrete glucagon and are involved in blood glucose elevation, into insulin-secreting β-cells has the potential to dramatically improve blood glucose levels.

Transdifferentiation into mature β-cells has been reported in many cell types including pancreatic ductal cells, hepatocytes, intestinal cells, as well as α-cells and δ-cells, which are adjacent to pancreatic β-cells within the pancreatic islets [50]. There are also reports on the transdifferentiation of α-cells to β-cells in vivo using a diabetic mouse model. A pancreatic β-cell-specific mutant K_ATP_ channel-expressing mice (with reduced responsiveness to ATP) exhibit pancreatic β-cell dedifferentiation in response to hyperglycemia, but when hyperglycemia is ameliorated by insulin administration to these mice, the number of Ngn3-positive cells decreases, and the number of insulin-positive cells increases [28]. In addition, when mice specifically expressing diphtheria toxin receptor (DTR) in their β-cells are administered diphtheria toxin, there is a rapid loss of β-cells, which causes the transdifferentiation from α-cells to β-cells [51]. In addition, a study using GLP-1-expressing adenovirus in mice reported that GLP-1 promoted the transdifferentiation from α-cells to β-cells via FGF21 [52]. This is supported by the fact that GLP-1 analogue liraglutide and dipeptidyl peptidase 4 (DPP-4) inhibitor sitagliptin suppressed the conversion of β-cells to α-cells in a diabetic mouse model [53]. In addition, administration of g-aminobutyric acid (GABA) to mice and humans also induced the transdifferentiation from α-cells to β-like cells. GABA supplementation to isolated human pancreatic islets results in the loss of α-cell mass and an increase in β-like cell mass [54]. In addition, administration of the antimalarial drug artemisinin to human pancreatic islets enhances GABA signaling and promotes the transdifferentiation from α-cells to β-like cells through the down-regulation of *Arx*, an α-cell-specific gene [55]. In this way, various substances induce the transdifferentiation from α-cells to β-cells in rodents as well as in human pancreatic islets.

The first report on the molecular mechanism of the transdifferentiation from α-cell to β-cell was an experiment using a glucagon-secreting cell line (αTC1-6). *Ins1* gene expression was induced by overexpression of Pdx-1 in αTC1-6 cells [56]. Similarly, ectopic Pax4 expression in glucagon-producing α-cells caused transdifferentiation to β-cells, resulting in an enlarged pancreas with β-cell hyperplasia [57]. Pax4-induced transdifferentiation from α-cells into β-cells was Ngn3-depedent [58]. Following this, Arx-specific inhibition in α-cells is involved in the conversion of α-cells to β-cells and Pax4 is essential for this process [59]. Furthermore, in adult mice, the simultaneous inactivation of DNA-methyltransferase 1 (Dnmt1) and Arx promotes the transdifferentiation from α-cells to β-like cells. The transformed β-like cells acquire electrophysiological functions characteristic of β-cells and exhibit GSIS [60]. In addition, it has been shown that MafA acts on Pdx-1 to promote the induction of β-cells from Ngn3-positive endocrine progenitor cells [61]. It has also been shown that α-cells were converted into functional β-cells when infected with adeno-associated virus (AAV), which induces the expression of Pdx-1 and MafA through the pancreatic duct [11]. In other words, in α cells, suppression of mature α-cell markers such as Arx and promotion of the expression of mature β-cell markers such as MafA and Pdx-1 may enhance the transdifferentiation from α-cells to β-cells. Thus, it may be possible to reciprocally convert cells in pancreatic islets using genetic manipulation in the future (Figure 3).

## 4. Potential Therapies Utilizing Differentiation and Transdifferentiation

Currently, the remedy for diabetes is symptomatic treatment. Thus, fundamental treatment is an ardent desire. Replacing lost β-cell mass and restoring its function is the ideal way to solve this problem. Additionally, the transdifferentiation from α-cells to β-cells as described above may lead to a cure for insulinopenic diabetes. The advantages of this approach are as follows: (1) differentiation and transdifferentiation of α-cells, which secrete glucagon and are involved in elevation of blood glucose, to β-cells, which secrete insulin and lower blood glucose, may dramatically improve blood glucose levels, (2) α-cells are embryologically similar to β-cells and have high potential for reprogramming, (3) both pancreatic α-cells and β-cells are located in the pancreatic islets, thus insulin is able to reach to liver directory, and (4) the proportion of α-cells is particularly high in human islets compared to that of rodents [62].

At this point in time, there is no doubt that improving hyperglycemia is the most useful means to protect and regenerate β-cells. As mentioned above, insulin therapy and improving blood glucose levels in diabetic model mice and human pancreatic islets lead to the protection of β-cells. Additionally, some β-cells are the result of transdifferentiation from α-cells. At the same time, in recent years, researchers have been looking towards gene therapy for diabetes. Specific expression of MafA, in the β-cells of obese type 2 diabetic model mice (db/db) recovered their ability to synthesize insulin, and improved GSIS accompanied with blood glucose amelioration [21]. In addition, β-cell-specific overexpression of Pdx-1 in diabetic mouse models restores β-cell function and improves glycemic control [19]. Furthermore, when adenoviral vector harboring both Pdx-1 and MafA mRNA was injected through the pancreatic duct, this resulted in Pdx-1 and MafA overexpression in pancreatic islets. This method induced α-cell to β-cell transdifferentiation either in type 1 or type 2 diabetic mice models, accompanied by amelioration of glucose intolerance [11]. Furthermore, administration of the viral vector to human pancreatic islets in which β-cells had been destroyed by streptozotocin (STZ) treatment induced transdifferentiation from α-cells to β-cells. They also reported that transplantation of these islets into non-obese diabetes/severe combined immunodeficiency (NOD/SCID) mice resulted in an improvement in blood glucose levels [11]. These strategies may enhance endogenous insulin secretion by increasing β-cell mass. This may be applicable to both type 1 and type 2 diabetes and is a curative treatment.

In addition, to date, pluripotent stem cells with the potential to multi-differentiate and self-renew have been intensively investigated as a strategy to generate mature pancreatic β-cells and islets. In 2014, stem-cell-derived β cells (SC-β) were generated from human stem cells (human induced pluripotent stem cells (hiPSC), human embryonic stem cells (hESC)), and these cells reportedly had a glucose-stimulated insulin secretion capacity equivalent to that of human mature β cells [63]. Additionally, transplantation of SC-β into diabetic mice was shown to improve hyperglycemia [63]. Thus, treatment using these pluripotent stem cells is expected to be the ultimate treatment for type 1 diabetes. On the other hand, induced pluripotent stem (iPS) cells are associated with a risk of carcinogenesis and embryonic stem cells are also associated with ethical issues, and the path to clinical application is far in the future.

Finally, the drugs that are currently under development and testing for pancreatic cell protection are listed below. Although tauroursodeoxycholic acid (TUDCA) had been approved for primary biliary cholangitis (PBC) by the Food and Drug Administration (FDA), it is also expected to reduce β-cell ER stress, thereby clinical studies are ongoing. TUDCA reduces ER stress by activating chaperone proteins. TUDCA has been shown to strongly inhibit the onset of diabetes in type 1 diabetes model mice [64]. The aforementioned antimalarial drug also promotes the transdifferentiation of pancreatic α-cells into pancreatic β-cells [55]. In addition, imeglimin, which will be launched worldwide between 2021 and 2022, is a mitochondrial-function-improving and insulin-resistance-improving agent [65]. Imeglimin activates mitochondrial complex III and contributes to the inhibition of ROS in pancreatic β-cells, that is, the reduction of oxidative stress and ER stress [65,66,67]. This causes protection of pancreatic β-cells and the inhibition of dedifferentiation can be expected. Thus, drugs that suppress pancreatic β-cell death and dedifferentiation and contribute to the protection of pancreatic β-cells have attracted attention.

## 5. Final Remarks

Until now, it has been thought that terminally differentiated β-cells end their life by apoptosis. However, in recent years, the concept of β-cell dedifferentiation has been proposed. It is now clear that β-cells dedifferentiate into immature cells due to metabolic stress and various other factors including hyperglycemia. In addition, these immature cells can transform into other islet cells once again. In recent years, it has been reported that transdifferentiation from other cells such as pancreatic α-cells to pancreatic β-cells are promoted to improve blood glucose in diabetes model mice. That is, dedifferentiation, transdifferentiation and redifferentiation of pancreatic β-cells are attracting more and more attention. Additionally, it is highly probable that up-regulation and down-regulation of transcription factors specifically expressed in terminally differentiated cells such as β-cells and α-cells are involved in these processes. Since pancreatic islet cells are originally differentiated from pancreatic endocrine progenitor cells, there is a possibility that pancreatic islet cells can transdifferentiate with each other.

## Figures and Tables

**Figure 1 metabolites-11-00218-f001:**
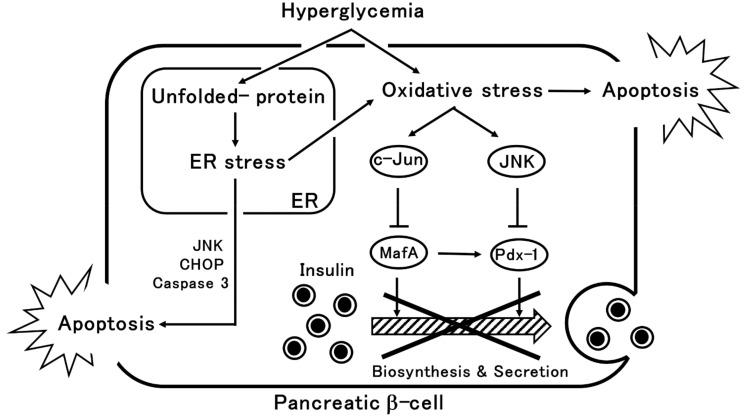
Mechanisms of pancreatic β-cell damage due to hyperglycemia. In chronic hyperglycemia, oxidative stress induces a decrease in the expression of MafA and Pdx-1, which are important transcription factors for the regulation of insulin gene expression, through c-Jun and JNK, resulting in decreased insulin biosynthesis and secretion. Oxidative stress induces apoptosis. Chronic hyperglycemia induces ER stress through the accumulation of unfolded proteins, leading to the apoptosis of pancreatic β-cells. Arrows indicate promotion and T arrows indicate inhibition.

**Figure 2 metabolites-11-00218-f002:**
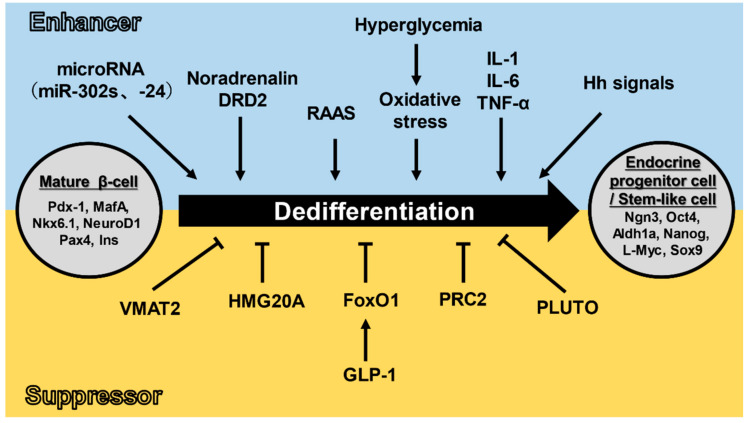
Mechanisms of dedifferentiation of pancreatic β-cells. Hyperglycemia, inflammatory cytokines (IL-1, IL-6, TNF-α), Hh signaling, RAAS, miRNA, noradrenaline, and DRD2 have been reported to enhance β-cell dedifferentiation. Molecules/peptides such as FoxO1, PRC2, HMG20A, PLUTO, VMAT2, and GLP-1 have been reported to suppress β-cell dedifferentiation. Arrows indicate promotion and T arrows indicate inhibition.

**Figure 3 metabolites-11-00218-f003:**
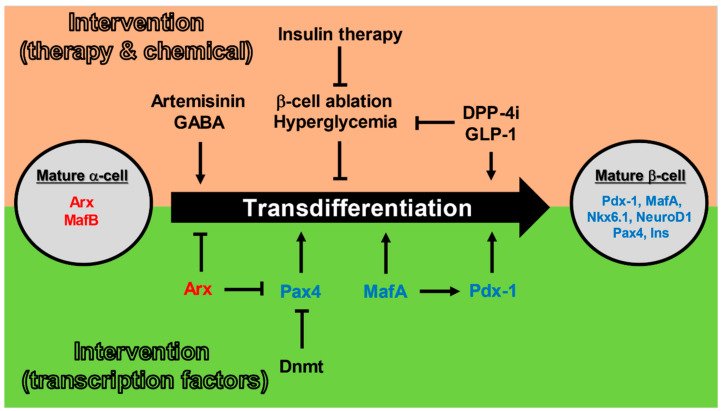
Mechanisms of transdifferentiation from pancreatic α-cells to pancreatic β-cells. Therapeutic interventions with insulin, GLP-1, or DPP-4i, as well as administration of GABA or artemisinin may promote the conversion of α-cells to β-cells. Pax4, MafA, and Pdx-1 (which are mature β-cell markers) may promote the conversion of α-cells to β-cells, while Arx and Dnmt may inhibit the conversion of α-cells to β-cells. Arrows indicate promotion and T arrows indicate inhibition.

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
