# Peer review of "The Plasticity of Pancreatic β-Cells"

_metabolites, 2021, doi:10.3390/metabo11040218_

Round 1

Reviewer 1 Report

This is an interesting article that focuses on an issue of great relevance in the comprehensive searching for potential treatment of preserve pancreatic β‐cell function. 

 However, it requires an in-depth review.

  1. Diabetes mellitus are protein conformational diseases (PCD) and the paradigm of non‐communicable diseases (NCDs) is shifting; the relevant literature need to be acknowledged to put this current study in perspective.
  2. This review is limited and it is not a systematic review that follow prisma protocol with a clear target question and objectives; is missing many relevant literature, (e.g. speak about the functional changes in β‐cell during ageing and senescence, chaperones genes, etc)

Reviewer 2 Report

This review article entitled," The plasticity of pancreatic β-cells' is quite informative and done with the diligent literature search. Herein authors have nicely-investigated the major signaling molecules in the transformation of pancreatic α and β- cell islets under hyperglycemia as well as elaborated the role of hyperglycemia-induced redox imbalance in differentiation and transdifferentiation of pancreatic β-cells.  This manuscript could be a valuable contribution to the readers interested in learning about the mechanistic of β-cells plasticity. However, the Authors need to describe the significance of Pluripotent stem cells and discuss some recent findings that showed cell differentiation and transdifferentiation from its lineage to give rise to a different cell type? 

β-Cell plasticity is one of the major reasons for failure in modern anti-diabetic drugs (especially insulinotropic drugs). Authors need to list current drug development to overcome β-cell dysfunction that prevents differentiation of cells.

Authors have suitably highlighted signaling pathways that instruct/regulate cells to differentiate into mature and functional β-cells and their differentiation into progenitor cells. However, the authors need to discuss how these signaling molecules can be a potential candidate for therapy and emphasized major driving factors in β-cell function, which need to be highlighted.   
